# Ultra-Light and Ultra-Low Thermal Conductivity of Elastic Silica Nanofibrous Aerogel with TiO2 Opacifier Particles as Filler

**DOI:** 10.3390/nano12223928

**Published:** 2022-11-08

**Authors:** Lixia Yang, Yang Ding, Mengmeng Yang, Yapeng Wang, Deniz Eren Erişen, Zhaofeng Chen, Qiong Wu, Guiyuan Zheng

**Affiliations:** 1School of Materials Science and Technology, Nanjing University of Aeronautics and Astronautics, Nanjing 211106, China; 2State Key Laboratory of Solidification Processing, Northwestern Polytechnical University, Xi’an 710072, China

**Keywords:** titaium oxide opacifier, electrospinning, elastic ceramic nanofibers, thermal conductivity, anti-infrared radiation

## Abstract

The thermal radiation phenomenon is more crucial than other thermal transportation phenomena at elevated temperatures (>300 °C). Therefore, infrared radiation resistance and its performance on thermal conduction of nanofibrous aerogel with Titanium oxide (TiO_2_) filler have been investigated compared to control groups (silica nanofibrous aerogels with and without filler). Nanofibrous aerogel has been produced by electrospun silica nanofibers. Later, TiO_2_ opacifier and a non-opacifier filled materials were prepared by a solution homogenization method and then freeze-dried to obtain particle-filled nanofibrous aerogel. Moreover, the thermal radiation conductivity of the composite was calculated by numerical simulation, and the effect of the anti-infrared radiation of the TiO_2_ opacifier was obtained. The fascinating inhibited infrared radiation transmission performance (infrared transmittance ~67% at 3 μm) and excellent thermal insulation effect (thermal conductivity of 0.019 Wm^−1^K^−1^ at room temperature) and maximum compressive strengths (3.22 kPa) of silica nanofibrous aerogel with TiO_2_ opacifier were verified. Excellent thermal insulation, compression and thermal stability properties show its potential for practical application in industrial production. The successful synthesis of this material may shed light on the development of other insulative ceramic aerogels.

## 1. Introduction

As the aerospace field has developed rapidly, the demand for thermal protection and energy conservation has increased dramatically [1,2,3,4,5,6]. For example, when an aircraft flies at high speed for a long period of time, the fuselage can be rubbed against a great deal of air resistance, producing a lot of heat and transfer to the interior of the spacecraft, causing great damage to pilots and instruments. Therefore, it is important to develop materials with high thermal insulation performance for safe flying.

The effective thermal conductivity of phenolic foam under high vacuum (<1 Pa) was determined and the result was only 0.005 Wm^−1^K^−1^, indicating that phenolic foam has excellent thermal insulation properties [7]. However, the phenolic foam has low elongation, a fragile structure, high hardness, and cannot be bent, so its application in the field of aerospace insulation is limited [8]. Ceramic aerogels have a complex structure with 3D network skeleton, which consists of interconnected spherical nanoparticles and randomly distributed micro or mesoscale pores [9].Ceramic aerogels are considered to be the solid material with the lowest thermal conductivity, meager solid volume ratio, and tortuous complex three-dimensional porous network structure, which significantly inhibits the solid-phase heat conduction of the material [10]. Generally, though, the inefficient structure continuity and connection result in poor mechanical properties [11,12,13,14]. Moreover, ceramic aerogels are usually rigid and brittle, with only a slight elastic deformation before fracture [15].

With the advancement of electrospinning technology, ceramic nanofibrous aerogel materials have become the mainstream of current research in the insulation field due to their flexibility, excellent chemical stability, and high-temperature resistance, which not only contribute to improving the ductility of the materials but also high-temperature thermal stability [16,17,18,19,20]. However, the temperature of a spacecraft during operation usually reaches the high-temperature range (>300 °C). For the reason that the radiative thermal conductivity is proportional to the third power of the temperature, neither the lowered density nor decreased infrared radiation absorption coefficient of ceramic nanofibrous aerogels are capable of preventing the thermal radiation heat transfer effectively [21,22,23,24,25]. It has been reported that the radiative thermal conductivity is reduced by coating the fiber surface with an opacifier. However, the infrared radiation still propagates through the large pores between the fibers, affecting the thermal insulation effect of the material [26]. On this point, preparing silica fibrous aerogels with excellent infrared extinction properties and low thermal conductivity at elevated temperatures is still extremely challenging.

Opacifier particles with high absorption mechanism or reflectivity for near-infrared bandwidth, such as Titanium oxide (TiO_2_) [27,28], carbon black [29], and SiC [30,31,32], were added to the aerogels as a filler to decrease the infrared transmittance significantly, thereby reducing radiative thermal conductivity at elevated temperatures [24,27,29,33]. Notably, the opacifier takes the volume fraction in a small proportion, the high porosity of nanofibrous aerogel can be maintained, and the pores are filled, which is beneficial to lower thermal conductivity [34,35,36]. The conductive thermal conductivity of the thermal insulation material mainly depends on the shape, size, and interconnection of the pores inside the material; the smaller the pore size, and the lower the thermal conductivity of the material [37,38]. The main reasons are (1) the minimization of the pore size reduces the convective heat transfer efficiency, and (2) the increase in the gas-solid interface raises the solid conduction distance and reduces the thermal conduction of the material [39]. Inspired by this point of view, a method of filling the pores of nanofibrous aerogels with opacifier particles was proposed.

In this study, nanofibrous aerogels with opacifier composites were prepared. The high-temperature insulation was designed based on two major criteria: (1) embedding the opacifier in the pores of the nanofibrous aerogel to shrink the pore size and decrease the thermal conductivity of the gas phase, and (2) the opacifier should have a high infrared radiation inhibition effect to control high-temperature radiation heat conduction. Silica nanofibrous aerogel with TiO_2_ particles (SNFAT) and two control samples, nanofibrous silica aerogel with SiO_2_ particles (SNFAS) and pristine silica nanofibrous aerogel (SNFA), were prepared. The thermal insulation performance of SNFAT, SANFAS, and SNFA, so nanofibrous aerogels with opacifier filler, non-opacifier filler, and pristine aerogel were investigated at elevated temperatures to investigate the effect of opacifier clearly. The structural variation and intrinsic mechanisms of nanofibrous aerogels with opacifier filler during compression were discussed, and their practical application was evaluated. Based on the improved thermal conductivity analysis model of nanofibrous aerogel with an opacifier, the radiative thermal conductivity of the samples was calculated, and the microscopic heat transfer mechanism of the aerogels was analyzed.

## 2. Experimental Procedure

### 2.1. Preparation of TiO_2_ Particles

The TiO_2_ nanoparticles were prepared by a sol-gel method. Tetrabutyl orthotitanate (TBOT), acetic acid (HAc), and anhydrous ethanol (EtOH) were used as starting materials. The procedure for the preparation of the particles is described as follows. Firstly, TBOT and EtOH were mixed and stirred uniformly according to the mass ratio of 1:3.5 to obtain solution A. Meanwhile, the H_2_O, HAc and EtOH were uniformly stirred according to the mass ratio of 1:1.2:16 to obtain solution B; then, the solution B was slowly dropped into solution A with stirring; then, the hydrolysis reaction was generated. After the procedure, formamide as drying control chemical additives was added, and the mass ratio of formamide to H_2_O was 0.07:1. The solution was kept at room temperature for 8 h to form a TiO_2_ gel, which is white and opaque; the gel was further put in of the EtOH solution for solvent exchange 2–3 times (8–12 h each time). At the end, the aged gel was dried successively at 60 °C, 80 °C, 100 °C, and 120 °C. All reagents were purchased from Sinopharm Chemical Reagent Corporation (China). 

### 2.2. Preparation of SiO_2_ Particles

The SiO_2_ particles were prepared in the same way as the TiO_2_ particles, but with the different precursors. Tetraethyl orthosilicate (TEOS), EtOH, water, and phosphoric acid (PPA) were mixed under vigorous stirring for 90 min. The mass ratio of TEOS:H_2_O:EtOH:NH_3_ H_2_O:PPA was fixed at 1:5:5:0.002:0.003. After sufficient hydrolysis of TEOS, NH_3_·H_2_O was added to the obtained sol under stirring for 10 min for the condensation reaction, and after a period of time to obtain the SiO_2_ gel, which is transparent and different from TiO_2_ sol. Aging for two days at room temperature after gelation. Finally, the aged gel was dried at 60, 80, and 100 °C for 9 h in a furnace to finally obtain the SiO_2_ particles.

### 2.3. Silica Nanofibers Prepared by Electrospinning

The flexible silica nanofibers were prepared using electrospinning and sol-gel methods, as illustrated in Figure 1a. The typical procedure is as briefly as follows: the silica precursor sol solution was prepared by stirring and mixing tetraethyl orthosilicate (TEOS), PPA, and 4 wt.% PVA with a mass ratio of 1:0.0125:1.25 at room temperature for 12 h. Following this process, the electrospinning process was performed with an applied high voltage of 15 kV and a constant feed rate of 0.5 mL h^−1^. The as-spun composite PVA/TEOS nanofibers were collected on the surface of a grounded aluminum foil-covered metallic rotating roller. Finally, the composite nanofibers were dried in an oven (60 °C) for 6 h and calcined at 800 °C in air by gradually increasing the temperature at a heating rate of 5 °C min^−1^ to obtain pure silica nanofibers, which exhibited average nanofibers diameter of 600 nm, as shown in Figure 1b.

### 2.4. Preparation Process of Nanofibrous Aerogels with Particles

The preparation process is shown in Figure 1c. SiO_2_ nanofibers membrane was cut into small pieces and soaked in solution 0.01 wt.% PAM solution. Later few drops of TEOS:H_2_O:EtOH:PPA = 1:4:10:0.01 solution were added into the solution to sustain viscosity. As filler materials with 70 wt.% of silica (for SNFAS samples) or TiO_2_ (for SNFAT samples) were added while a control group (SNFA) was kept pristine, then the mixed solution was dispersed and homogenized by a high-speed homogenizer. In the freezing procedure, the homogenized nanofibers dispersions were transferred to the desired mold and then frozen in a liquid nitrogen bath. During this process, the ice crystals in the solution rapidly nucleated and grew. After the solvent in the dispersion was completely solidified, a frozen block was obtained. Then, the frozen block was placed in a freeze dryer, and the solvent was directly sublimated from a solid state to a gaseous state by controlled temperature and pressure (−50 °C and 0.05 Pa) to remove the solvent ice crystals without destroying the pore structure of the material to obtain nanofibrous aerogel with particles [40].

### 2.5. Characterizations

Morphology and microstructure were characterized by a scanning electron microscope (LYRA3GMH, Brno, Czech Republic). X-ray diffraction (XRD) patterns were recorded with a smart X-ray diffractometer (Empyrean, Almelo, The Netherlands). Thermal conductivity was measured using a thermal conductivity tester (Hot Disk TPS 2500S, Uppsala, Sweden), the samples size was 25 × 25 × 3 mm and the sensor was Kapton 5465. The compressive stress-strain curve was measured by a universal mechanical testing machine (CMT6103, Shenzhen, China). Fourier infrared spectroscopy (Nicolet iS20, Waltham, Mass, USA) was used to characterize the infrared transmittance; in order to collect the Fourier spectrum of the samples, potassium bromide powder was mixed with the samples and pressed into sheets. Thermographic images were taken with an infrared thermal camera (Fotric 310, Shanghai, China). Thermal analysis was conducted with Thermogravimetric analysis (TGA STA449F5, Selb, Germany) in air. The pore size distribution was measured by automated surface area and porosity analyzer (ASAP 2460, Norcross, Georgia, USA). The density was calculated from the size and mass of the sample.

## 3. Results and Discussion

### 3.1. Micromorphology of Nanofibrous Aerogels with Particles

The mechanical and thermal properties of nanofibrous aerogel materials are usually closely related to the morphology and structure of nanofibrous membranes [15,21,41]. The mechanical properties of silica nanofibrous membranes prepared by electrospinning differ from conventional brittle and hard bulk silica-based materials include conventional aerogels [42]. The electrospun silica nanofibrous membrane has elasticity then can be restored to its original shape after being folded repeatedly. The nanofibers did not crack or break after being bent at a large angle, which can be observed in Figure 1b.

The optical image of the silica nanofibrous aerogels with particles prepared by the homogenization method is shown in Figure 1c. According to the SEM images (Figure 2), the silica nanofibrous aerogels with particles can be divided into three parts at different scales [43], single cell (50–100 μm, Figure 2a,d,g), cell walls (1–10 μm, Figure 2b,e,h), silica nanofibers (300–600 nm, Figure 2c,f,i). The silica nanofibrous aerogels with particles take on an obvious hierarchical structure, with unidirectionally arranged long pore channels in the cross-section, with a high degree of regularity, similar to a porous honeycomb structure; yellow markers indicated easier-to-find information in Figure 2. The reason for the formation of the hierarchical structure was that the homogenized solution was frozen in liquid nitrogen, and ice crystals grow from the bottom of the solution, displacing the nanofibers and particles around. After freeze-drying, the ice crystals are sublimated and maintain their original shape [44]. The preparation of flexible nanofibrous aerogels with particles by homogenization also possess a unique advantage; they can be prepared into any shape and can form a hierarchical structure, which is beneficial for some applications that require a specific shape. Strong mechanical support is provided by unique interwoven nanofibers in the cell walls (Figure 2b,e,h), and high strength and compressible recovery properties are endowed to the aerogels. Similar results have been reported by Dou et al. [17]. The pore size between nanofibers is shrunk by the micron-sized opacifier particles attached to the nanofibers walls (Figure 2b,h), which is very effective in reducing heat transfer [27].

The distribution of TiO_2_ opacifier particles in the silica nanofibers was further supported by X-ray spectroscopy (EDS) mapping results, as illustrated in Figure 3a–d, Ti element was randomly filled between Si and O elements in SNFAT, which indicated the coexistence of TiO_2_ opacifier and nanofibers. Moreover, the brighter, clustered green spots in Figure 3c indicated the presence of TiO_2_ opacifier particles at this location. According to the range of the spots, it can be known that the size of the sunscreen particles is several to several tens of micrometers.

The microstructure of SiO_2_ and TiO_2_ particles were observed by transmission electron microscopy (TEM), and the results are shown in Figure 4. The TEM image of the selected area in Figure 4a,b,d,e revealed that both SiO_2_ and TiO_2_ particles have a dense structure. Figure 4c,f further showed the amorphous structure of SiO_2_ and TiO_2_, indicating the amorphous structure of silica nanofibrous aerogels with particles. The same method was used by Thangavel et al. [45] and Gao [46] to characterize the amorphous structure of SiO_2_ and TiO_2_, respectively, and the results were the same as in this study.

To gain insight into the phase structure, the diffraction peak information of the samples was observed by XRD. As shown in Figure 5, SNFAT had no diffraction peak, hence, as expected, the SNFAT stayed amorphous, similar study has been announced by Keri et al. [47]. SNFA and SNFAS showed a wide peak around 22°, which revealed the typical amorphous structure and no phase transition occurs [48]. XRD patterns was in good agreement with TEM analysis.

### 3.2. Mechanical Properties

Mechanical properties are essential for practical application of high-temperature thermally insulating nanofibrous aerogels [49]. The mechanical properties of nanofibrous aerogels with particles were investigated (Figure 6). As shown in Figure 6, nanofibrous aerogels with particles were light enough to stand on a dandelion easily, which can be verified by their densities.

Compared with conventional silica aerogels, SNFAT, SNFAS, and SNFA all exhibited excellent flexibility without fracture and irreversible deformation under larger compression. Stress-strain (σ-ε) curves for a set strain of 40%, which does not cause plastic deformation of the samples [17], can be observed in Figure 7, and the three samples can fully recover below ε < 40%. The stress loading and unloading process was illustrated at the red allow in the Figure 7. The Similar to the compressive states of other fibers aerogels [16,50], two states are clearly observed in the 40% stress-strain (σ-ε) curve: ε < 20% exhibits a linear elastic state with a reduced slope of the compression curve. In the subsequent plateau period of 20% < ε < 40%, stable elastic deformation will occur at this stage. The maximum compressive strengths of SNFAT, SNFAS and SNFA are 3.22 kPa, 10.27 kPa, and 6.71 kPa, respectively, and the samples do not exhibit brittle fracture characteristics which shows that the nanofibers have a toughening effect.

This compressive elasticity can be attributed to the following reasons: (1) One-dimensional flexible silica nanofibers are endowed with a fundamental toughness that can be attributed to the amorphous structure of silica that can trigger shear bands to improve deformation performance, which is key prerequisites for bendable properties of ceramic nanofibers [17,51]; (2) the connected nanofibers network reduces the plastic deformation induced by the dislocations between adjacent nanofibers to improve the elasticity and reduced the residual strain, which is the reason why the compressive strength of SNFAT is lower than that of SNFAS and SNFA [21]; (3) structural robustness and elasticity are provided by the hierarchical cells structure, and the ordered arrangement of long pore channels means that the nanofibers cell walls do not touch each other to provide lateral expansion force, thus compressive strain is absorbed by the buckling and inversion of the nanofibers cells. A similar study has been announced by Si et al. [17].

### 3.3. Thermal Properties

Low thermal conductivity materials are used to lower energy loss and thermal insulation with important applications in the aerospace industry [21]. As shown in Figure 8 the SNFAT, SNFA and SNFAS showed low thermal conductivities of 0.019 Wm^−1^K^−1^, 0.022 Wm^−1^K^−1^, and 0.025 Wm^−1^K^−1^, respectively, which are lower than the thermal conductivity of micron fiber aerogels (~0.047 Wm^−1^K^−1^) [27]. It was worth noting that SNFAT exhibited lower thermal conductivity than SNFA and SNFAS. At room temperature, the thermal conductivity of aerogels usually consists of three parts: solid-phase thermal conductivity, gas-phase thermal conductivity and radiative conductivity [52,53]. The solid-phase heat conduction of ceramic nanofibrous aerogels fundamentally differ from conventional ceramic-based bulk aerogels. With the same volume, curved nanofibers can provide longer propagation paths for solid heat conduction. Amorphous silica nanofibers with low thermal conductivity are used as the base material, and the interconnected layered structure of nanofibers provides infinite heat conduction paths, which can effectively reduce the solid-phase thermal conductivity [54]. However, considering that the solid-phase thermal conductivity of TiO_2_ particles is slightly higher than that of SiO_2_ particles, the reason for the difference in thermal conductivity among the three samples is not the solid-phase thermal conductivity.

The pore size distribution of the samples was shown in Figure 9; most of the pore sizes were concentrated in 5–20 nm. The size of mesopores are even smaller than the mean free path of air molecules, which has more inhibitory effect on gas heat flow [55]. Notably, the number of the mesopores located in 5–10 nm of the sample SNFAT were much more than that of the samples SNFA and SNFAS, so the gas-phase thermal conductivity of SNFAT should be much lower than the control samples. Interestingly, we found that the radiative thermal conductivity at room temperature also has a non-negligible contribution to the heat conduction, which will be proved in Section 3.4.

At high temperature, the thermal conductivity of aerogels is mainly radiative heat transfer. In order to further study the thermal insulation properties of SNFAT, SNFA and SNFAS, dynamic distribution of the temperature heated by the flame was observed by an infrared thermal imager, as shown in Figure 10. The heating plate was heated on the alcohol lamp for 10 min in advance to ensure the constant temperature of the heating plate in the subsequent heating process (Figure 10b); the heating process was shown in Figure 10c. SNFAT, SNFA and SNFAS with a thickness of 2.5 cm were placed on a heating plate at 475.2 ± 4 °C for 10 min (Figure 10d,e). It can be clearly seen that the nanofibrous aerogel with particles has different temperature layer distributions, the height of the red high temperature layer of SNFAT is nearly 50% lower than that of SNFAS and SNFA, and the temperature of the cold end is 43.3 °C, 55.5 °C, 57.7 °C, respectively (Figure 10c–e). This data has proved that SNFAT, SNFA and SNFAS have excellent high-temperature thermal insulation properties, and that the thermal insulation effect of SNFAT was particularly outstanding, which probably due to the good infrared radiation suppression effect of TiO_2_.

Wien’s displacement law can measure the relationship between the temperature and infrared wavelength of radiation heat transfer of matter, which can be expressed as:(1)$$T\lambda =2.9\times 10^{-3}$$
where *T* is the temperature, *λ* is the wavelength, and 2.9 × 10^−3^ is the Wien constant.

The mid-infrared wavelengths of 3–6 μm correspond to the peak wavelengths in the main region of 200–700 °C. In order to verify the infrared suppression effect of TiO_2_ opacifier particles, FT-IR spectra of three aerogels were measured. The infrared transmittance spectra of the three samples were shown in Figure 11a. The infrared transmittance (>85%) of SNFA is extremely high at 3–6 μm, and infrared radiation can easily pass through at 200–700 °C. Compared with SNFAS, the infrared transmittance of SNFAT is significantly decreased, which indicates that the contribution of TiO_2_ particles to blocking thermal radiation is much greater than that of SiO_2_ particles. It reflects that the thermal insulation properties of the composites are improved by adding TiO_2_ particles in the nanofibrous aerogel. FT-IR spectra of samples heat-treatment at 800 °C were also measured to exclude the influence of organic matter on infrared transmittance. As shown in Figure 11b, the sample after heat treatment and the sample before heat treatment keep the same rule at 3–6 μm; SNFAT has the lowest infrared transmittance.

Thermogravimetric (TG) curve reflected the transformation process of the weight of three nanofibers aerogels. As shown in Figure 12, the TG curves of SNFA and SNFAS have three different stages [56]. In stage A (20–250 °C), the residual solvent and adsorbed water are evaporated, the samples weight were reduced by 1.69% and 7.21%, respectively. Then, in stage B (250~480 °C), the decomposition of PAM caused the weight loss of 3.25% and 3.53%. The side chains if polymer started to break and the inorganic transformation was completed in stage C (480~800 °C). The stage C of SNFAT was divided into two parts (C (1) and C (2)). The weight loss of Stage C (1) may be caused by the transition of titanium dioxide from amorphous precipitate to anatase phase [57]. The stage C (2) was identical to the stage C of SNFA and SNFAS.

Thermal stability of nanofibrous aerogels with particles was evaluated. As shown in Figure 13a–i, on a heating plate with a temperature of about 450 °C, nanofibrous aerogel is compressed and deformed by ~50% and can be quickly restored to its original shape, no melting, cracking, and structural collapse were observed, highlighting the temperature-invariant elasticity. Significantly, the effect of dense nanopores on thermal insulation is reflected in the thermal conductivity. To demonstrate the superior insulation performance of SNFAT, the density and thermal conductivity of porous insulation materials was compared with SNFAT. The results can be observed from Figure 14, SNFAT exhibits much lower density and thermal conductivity compared with different kinds of fibrous aerogels, which is very promising for practical lightweight thermal insulation applications [17,58,59,60,61].

### 3.4. Numerical Simulation of Radiative Thermal Conductivity

The radiative thermal conductivity *K_R_* can be expressed as [62]
(2)$$K_{R}=\frac{16n^{2}\sigma T^{3}}{3\beta_{R}}$$
where *n* represents the total refractive index of the composite material; *σ* represents the Stefan–Boltzmann constant, *σ* = 5.67 × 10^−8^ Wm^−2^K^−4^; *β_R_* represents the Rossland average extinction coefficient.

*β_R_* is a function related to the specific extinction coefficient and can be expressed as
(3)$$\beta_{R}=\beta \rho$$
where *β* is the specific extinction coefficient and *ρ* is the sample density.

The specific extinction coefficient is a temperature-dependent function that determines the ability of a opacifier to suppress infrared radiation at different temperatures, and in general the specific extinction coefficient is negatively related to radiant thermal conductivity [63]. The specific extinction coefficient is determined by the infrared transmittance of the sample
(4)$$\beta =-\frac{\ln(\tau )}{L\rho }$$
(5)$$L=\frac{Wm_{f}}{A_{1}\rho }$$
where *τ* is the infrared transmittance; *L* is the equivalent thickness of the potassium bromide sample after tableting; *W* is the total mass of the potassium bromide pellet; *m_f_* is the mass fraction of the sample powder in the potassium bromide pellet; and *A*_1_ is the cross-sectional area of the potassium bromide pellet.

When thermal radiation is projected on the surface of the opacifier, three phenomena generally occur, namely absorption (*Q_α_*), reflection (*Q_p_*) and penetration (*Q_r_*). Infrared opacifier has a large refraction and extinction coefficient, so the opacifier particles has excellent refractive and absorptive index, which reduces the heat transfer of thermal radiation and improves the thermal insulation performance of the materials [29].The mid-infrared wavelength of 3 μm corresponds to a higher actual temperature. The specific extinction coefficients of the three nanofibrous aerogels with particles were calculated using the above method, and the results are shown in the Figure 15a. The specific extinction coefficient curve of SNFAT in the range of 3–8 μm is above the other two samples, which indicated that the addition of TiO_2_ opacifier leads to a higher specific extinction coefficient, which is beneficial to reduce the radiative thermal conductivity of the samples. The calculation results in Figure 15b show that the radiative thermal conductivities of the three samples, which are small but still accounts for a certain proportion of the total thermal conductivity at room temperature. With the increase of temperature, the radiative thermal conductivity increases sharply and shows a large difference, among which the radiative thermal conductivity of SNFAT is about 4 times lower than that of SNFA at 600 °C, which can be attributed to the large complex refractive index of TiO_2_.

## 4. Conclusions

Elastic ceramic nanofibers with aerogel were successfully prepared by homogenization method and freeze-drying process with hierarchical porous structure, excellent anti-infrared radiation performance, low thermal conductivity, low density, and rapid recovery after compression. By comparison, the silica nanofibrous aerogel with TiO_2_ opacifier possess ultra-low thermal conductivity (0.019 Wm^−1^K^−1^ at room temperature) and excellent infrared radiation suppression performance at high temperature, and the thermal conductivity of the gas phase is significantly reduced by the smaller pores. These results shows that the embedded TiO_2_ opacifier enables nanofibrous aerogels to have excellent thermal insulation properties at both low and high temperatures. The numerical simulation results show that the TiO_2_ opacifier has a larger specific extinction coefficient at 3–6 μm, which reduces the infrared thermal radiation. This excellent performance makes our material promising and provides new insights into producing efficient thermal insulation materials that can be used in high temperatures and harsh environments.

## Figures and Tables

**Figure 1 nanomaterials-12-03928-f001:**
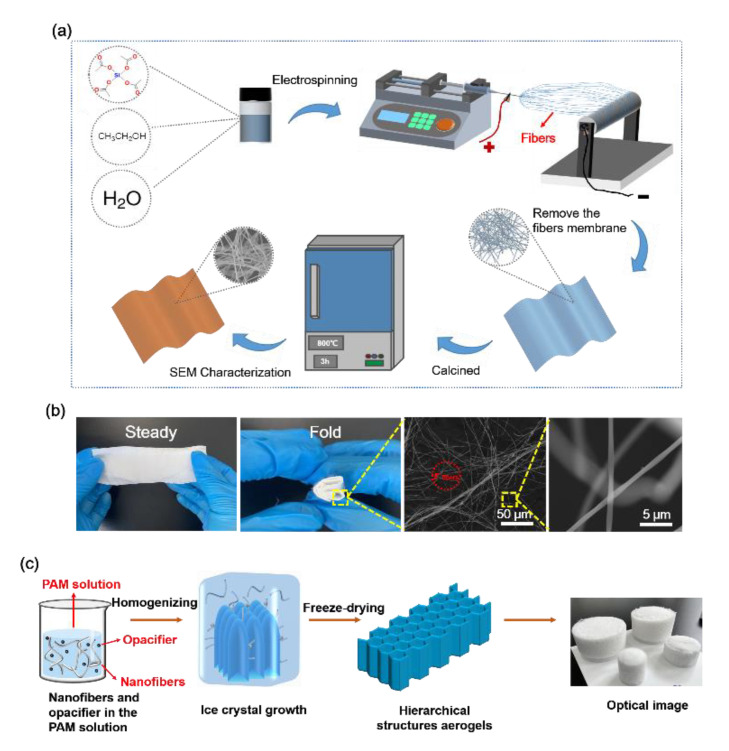
SiO_2_ nanofibrous aerogel with particles fabrication process (**a**) Preparation of ceramic nanofibers by electrospinning; (**b**) Optical images and SEM images of flexible SNF (silica nanofibers) membrane; (**c**) Fabrication process of nanofibers composite aerogels.

**Figure 2 nanomaterials-12-03928-f002:**
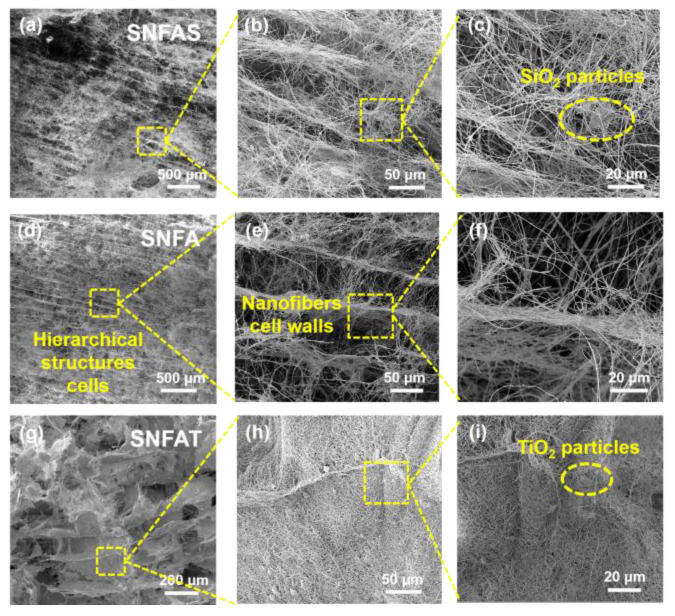
SEM images with hierarchical cells structures at different magnifications; (**a**–**c**) silica nanofibrous aerogel with SiO_2_ particles; (**d**–**f**) silica nanofibrous aerogel; (**g**–**i**) silica nanofibrous aerogel with TiO_2_ particles.

**Figure 3 nanomaterials-12-03928-f003:**
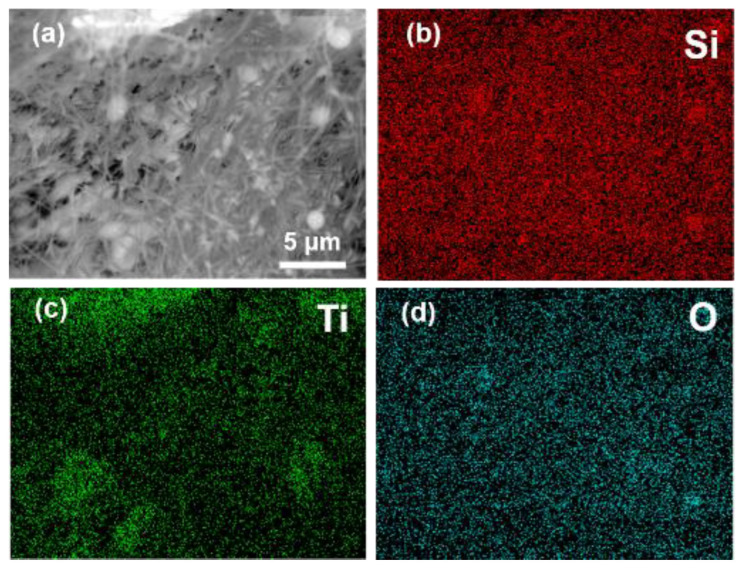
EDS images of (**a**–**d**) SNFAT and corresponding Si, Ti, O elemental mapping.

**Figure 4 nanomaterials-12-03928-f004:**
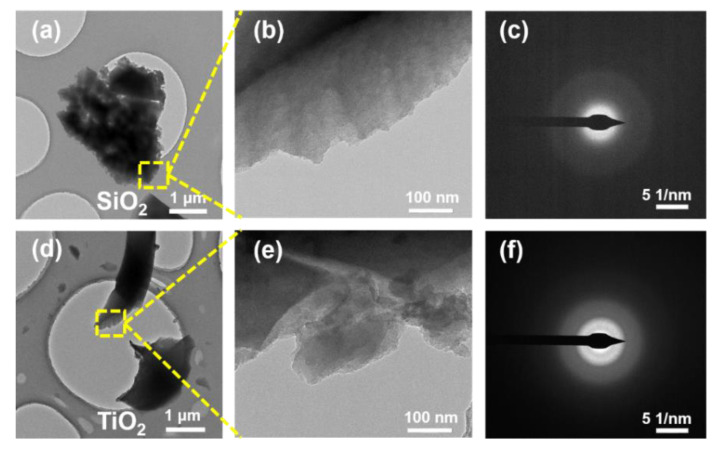
TEM images of (**a**–**c**) SiO_2_ and (**d**–**f**) TiO_2_ particles.

**Figure 5 nanomaterials-12-03928-f005:**
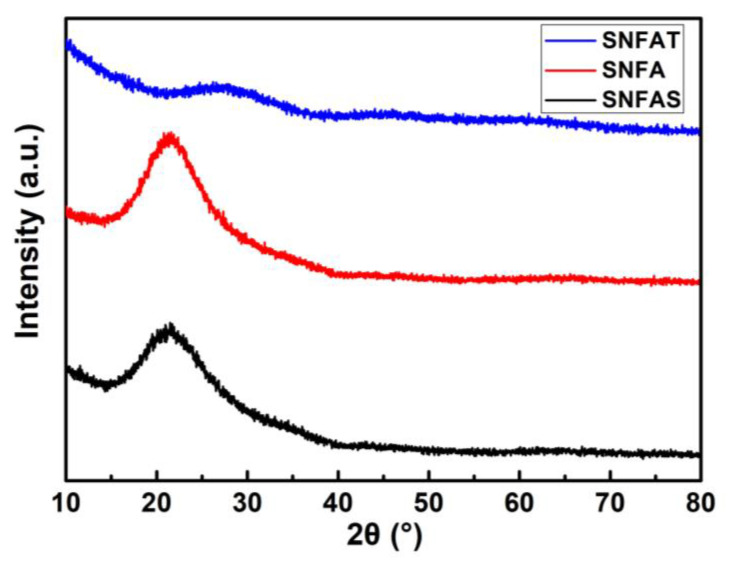
XRD patterns of SNFAT, SNFA and SNFAS.

**Figure 6 nanomaterials-12-03928-f006:**
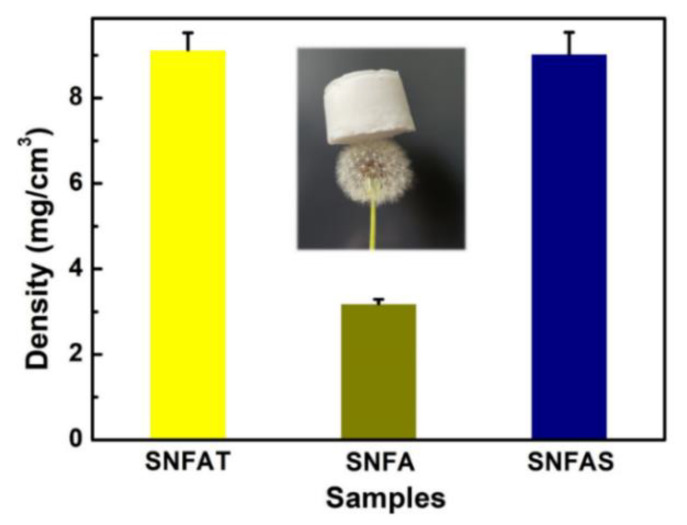
Density of SNFAT, SNFA and SNFAS.

**Figure 7 nanomaterials-12-03928-f007:**
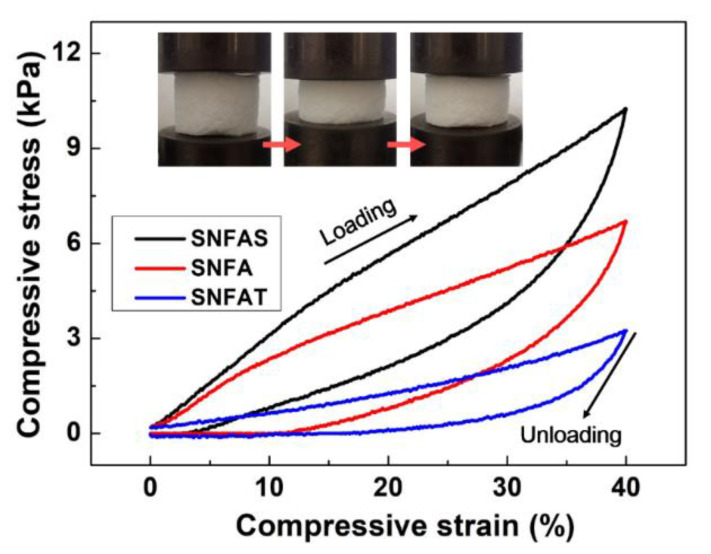
Stress-strain curve of the SNFAT, SNFA, and SNFAS at set strains of 40%.

**Figure 8 nanomaterials-12-03928-f008:**
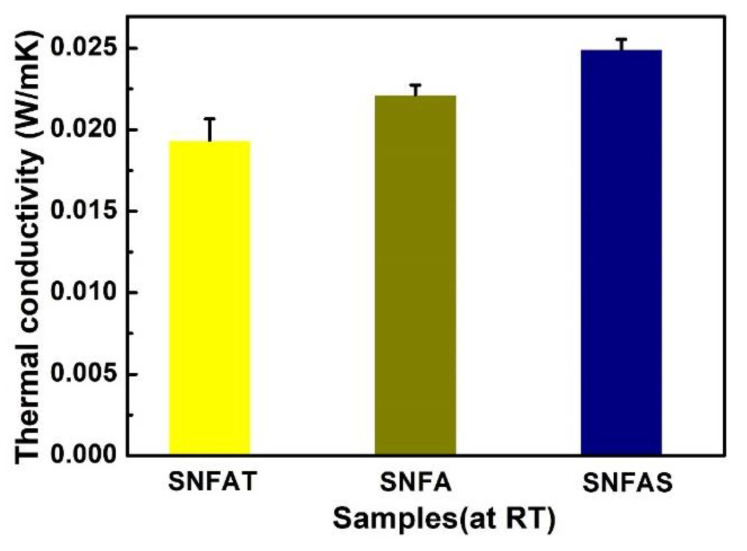
Thermal conductivity at room temperature of SNFAT, SNFA and SNFAS.

**Figure 9 nanomaterials-12-03928-f009:**
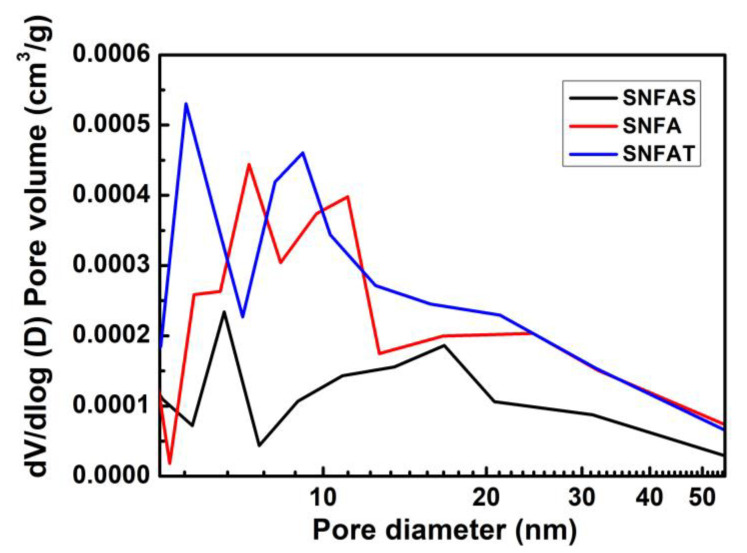
The pore size distribution of SNFAS, SNFA and SNFAT.

**Figure 10 nanomaterials-12-03928-f010:**
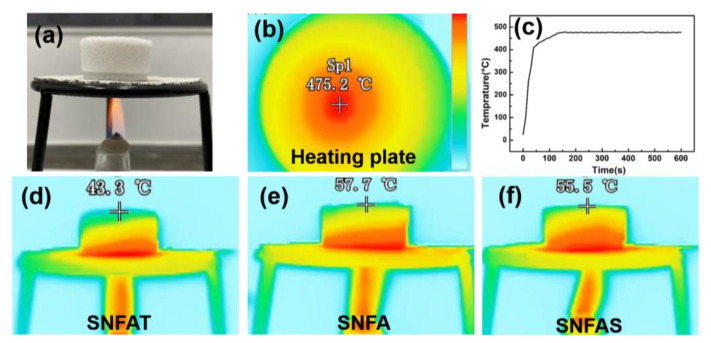
(**a**) Optical images of the heating process. (**b**) Infrared images of the heating plate. (**c**) Heating curve of heating plate for 10 min. (**d**–**f**) Infrared images of SNFAT, SNFA, and SNFAS on a heating stage at 475 °C for 10 min.

**Figure 11 nanomaterials-12-03928-f011:**
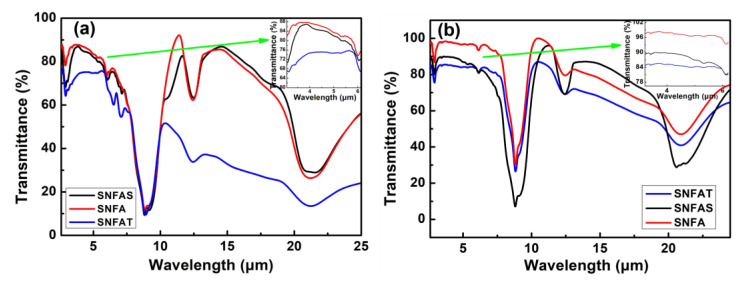
Infrared transmittance of SNFAS, SNFA and SNFAT. (**a**) No heat treatment. (**b**) Heat treatment at 800 °C.

**Figure 12 nanomaterials-12-03928-f012:**
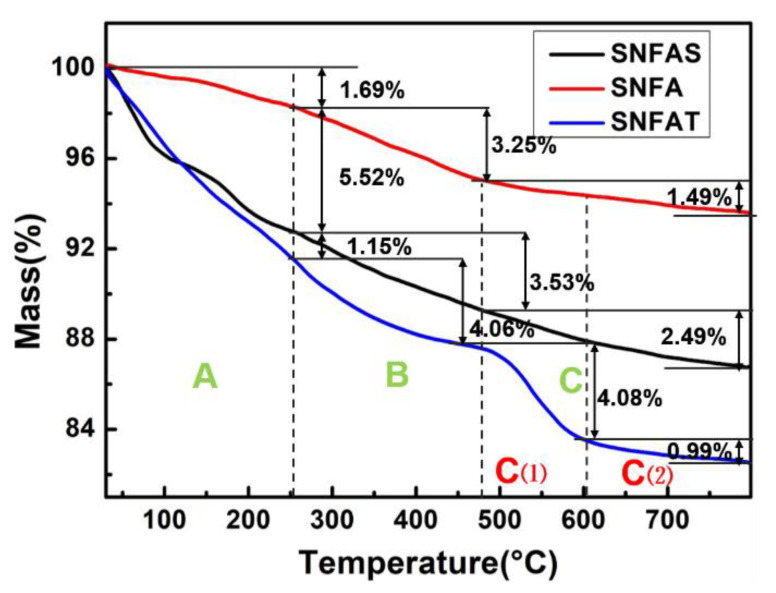
TG curve of SNFAS, SNFA and SNFAT (SNFAS and SNFA have three stage A, B and C, SNFAT has four stage A, B, C (1) and C (2)).

**Figure 13 nanomaterials-12-03928-f013:**
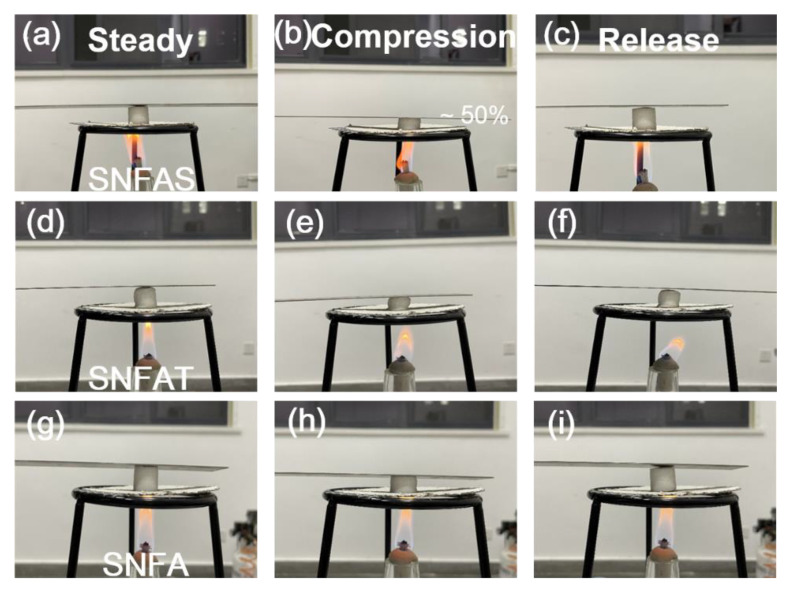
Compression properties of (**a**–**c**) SNFAT, (**d**–**f**) SNFA and (**g**–**i**) SNFAS at high temperatures.

**Figure 14 nanomaterials-12-03928-f014:**
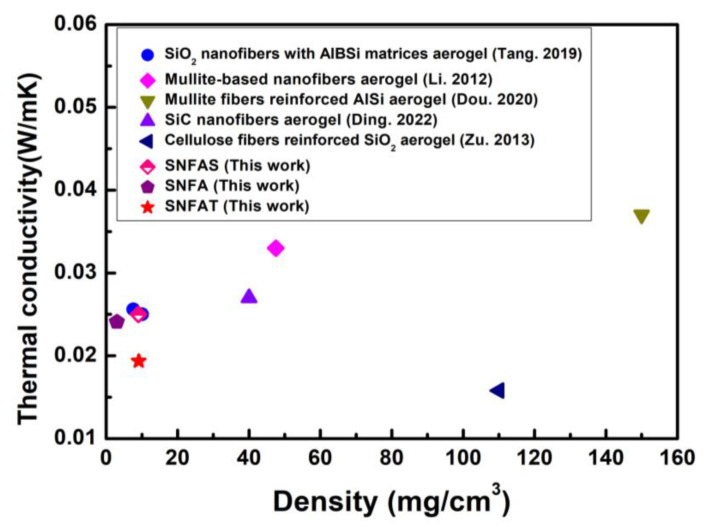
Thermal conductivity versus density for fibrous aerogel-like materials [8,39,40,41,42].

**Figure 15 nanomaterials-12-03928-f015:**
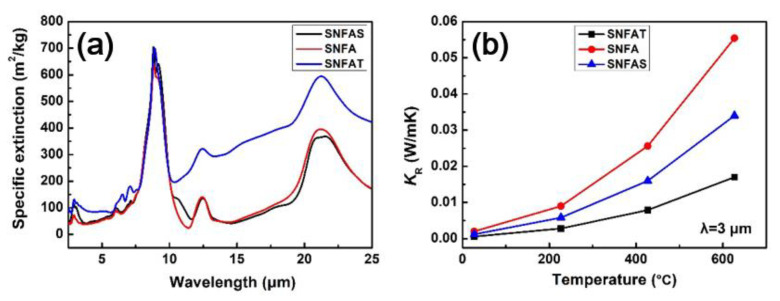
(**a**) The specific extinction coefficient of SNFAS, SNFA, and SNFAT; (**b**) Radiative thermal conductivity *K_R_* at 3 μm wavelength at different temperatures.

## Data Availability

The data presented in this study are available on request from the corresponding author.

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
