# Peer review of "Ultra-Light and Ultra-Low Thermal Conductivity of Elastic Silica Nanofibrous Aerogel with TiO2 Opacifier Particles as Filler"

_nanomaterials, 2022, doi:10.3390/nano12223928_

Round 1

Reviewer 1 Report

The paper sounds good and could be published with minor revisions. Some remarks R and questions Q are listed bellow:

R1 the authors pointed out, that the SiO2 and TiO2 preparation followed a similar procedure. Indeed, there are similarities, but also differences coming from the different precursors used here. A comparison together with a short coment would be interesting for the reader.

Q1 The TiO2 prepartion and optical properties here are similar to recently published results /Molecules 26 (2021), but the reference is not cited in the text. It would be interesting to compare both preparation ways and properties of TiO2 .

Q1 The value of lambda = 0.019 W/mK is very low, what did the author use as a reference for verification of such low values?

R1 The abstract need a shorter presentation, giving the results obtained here, without elements of an introduction.

Reviewer 2 Report

The manuscript “Ultra-light and ultra-low thermal conductivity of elastic silica nanofibrous aerogel with TiO2 opacifier particles as filler” reports on the modification of electrospun silica aerogels with TiO2 nanoparticles to lower their thermal conductivity. Ultra-low density aerogels with excellent thermal insulating properties in a wide range of temperatures were thus obtained. These aerogels also possessed excellent mechanical properties.

Although the results are quite promising, the study has some serious flaws: it lacks important material characterization, discussion of some major points and, most importantly, it lacks discussion of the thermal conductivity results. So, the manuscript should be re-considered after major revision.

1) I have serious concerns on the design of the thermal conductivity experiment and its discussion:

1.1.) The heat transfer experiments for the aerogel monoliths were conducted using an open flame of a simple alcohol burner and a thermal image camera (Fig. 9). How the hotplate temperature could be kept constant with the accuracy of one tenth degrees centigrade for 10 mins (line 268)? Please provide the measurement error for the temperature of the hotplate. Please provide also the data on the changes in the hotplate temperature during the experiments.

1.2) The discussion of the thermal conductivity (lines 240–261) refers to the SEM images with tens of microns scale bars. In this regard, the claims “nano-scale pores in the nanofibers wall” and “the size of nano-scale pores are even smaller than the mean free path of air molecules” seem to be unsupported with the data of the nano-pore sizes and the mean free path of air molecules. Please provide pore size distributions. Surface area values are also advisable.

2) Upon freeze-drying procedure, organic impurities remain in aerogels (PAM and PPA, alkoxy-groups) which will decompose upon heating up during the thermal conductivity studies. Such a decomposition should result e.g. in spatial differences of the thermal properties of the materials. Please, discuss this problem. FTIR spectra and TGA data for as-prepared aerogels should be also provided.

3) Please clarify the term “ceramic aerogels” in the Introduction section.

4) The XRD patterns for SNFA and SFNAS samples shows diffraction maxima at ~26, 48 and 17, 23, 28, 44 °2θ, respectively. These peaks are obviously due to the presence of crystalline phases that should be identified, and their formation should be discussed. In Fig. 5, the sample names should be corrected.

5) Please comment on the choice of the stress strain threshold (40%) for mechanical properties study.

6) Please discuss the mechanism of the suppression of IR irradiation by TiO2-modified aerogels. How TiO2 opacifier results in lower thermal conductivity?

7) Please increase contrast and brightness in EDX-mapping images (Fig.3). The signal can hardly be seen.

Reviewer 3 Report

This paper describes the production and characterization of nanofibrous aerogel with TiO2 opacifier particles as filler. Although interesting, there is a lot of missing information. Used characterization techniques must be better described, in addition, some conclusions cannot be extracted from the obtained data. I consider the manuscript needs major revision before accepting.

Typos:

-Line 37: “The effective thermal conductivity of phenolic foam under high vacuum (< 1Pa) and the result were only”. The verb is missing, The effective thermal conductivity … was determined (or measured) and the result …

- Formula in line 52 should be added as an equation and each symbol must be explained.

- Line 61: titanium oxide has been written in a different font.

-Line 131: “then broke the silica nano fibers membranes into homogeneously-dispersed nanofibers dispersions”. I do not understand this sentence, please explain it.

Minor modifications:

-        Please reference figure 1.a while you are explaining the electrospinning method.

-        Please add the pressure and temperate for the drying process (section 2.4)

-        Please add a reference of micron fiber aerogels thermal conductivity ( line 243)

-        Caption of figure 9 refers to image e that is missing.

Mayor:

-        The thermal conductivity method should be better explained in the experimental section. Dimension of the samples? Used sensor…?

-        In figure 2 particles are hardly distinguishable, please improve the contrast or the magnification of the images

-        Caption in Figure 3 does not correspond to the image.

-        In figure 3, I cannot distinguish the uniformity of Ti, please explain, or show it clearly.

-        Authors should explain the differences between the three materials regarding XRD.

-        The authors affirm that the SNFAT present a lower thermal conductivity due to the gas phase attributing this effect to the nano-scale pores in the nanofibers wall of SNFAT. They also claimed that the conduction through the solid phase is not different in the three materials.

o   Nano-scale pores are not shown in any Figure of the paper or are not distinguishable.

o   The obtained data is not enough for such affirmations. The authors should measure the thermal conductivity of the three materials' solid phase, which means fibers without the gas phase. This could be done by compacting the fibers leading to a material without porosity. Then the difference due to the added particles can be determined, and more conclusions can be extracted.

-        Line 327 suddenly authors start talking about potassium bromide pellet? 

Round 2

Reviewer 2 Report

The authors have addressed most of my comments. The paper is now suitable for publication.

Reviewer 3 Report

After the revisions, I consider the manuscript can be accepted.